# PrimDiffusion: Volumetric Primitives Diffusion for 3D Human Generation

**Zhaoxi Chen**[1] **Fangzhou Hong**[1] **Haiyi Mei**[2] **Guangcong Wang**[1]
**Lei Yang**[2] **Ziwei Liu**[1,✉]

[1]S-Lab, Nanyang Technological University [2]Sensetime Research

{zhaoxi001, fangzhou001, guangcong.wang, ziwei.liu}@ntu.edu.sg
{meihaiyi, yanglei}@sensetime.com

## Abstract

We present **PrimDiffusion**, the first diffusion-based framework for 3D human generation. Devising diffusion models for 3D human generation is difficult due to the intensive computational cost of 3D representations and the articulated topology of 3D humans. To tackle these challenges, our key insight is operating the denoising diffusion process directly on a set of volumetric primitives, which models the human body as a number of small volumes with radiance and kinematic information. This volumetric primitives representation marries the capacity of volumetric representations with the efficiency of primitive-based rendering. Our PrimDiffusion framework has three appealing properties: **1)** compact and expressive parameter space for the diffusion model, **2)** flexible 3D representation that incorporates human prior, and **3)** decoder-free rendering for efficient novel-view and novel-pose synthesis. Extensive experiments validate that PrimDiffusion outperforms state-of-the-art methods in 3D human generation. Notably, compared to GAN-based methods, our PrimDiffusion supports real-time rendering of high-quality 3D humans at a resolution of $512 \times 512$ once the denoising process is done. We also demonstrate the flexibility of our framework on training-free conditional generation such as texture transfer and 3D inpainting.

## 1 Introduction

Generating 3D humans with high-resolution and 3D consistency is a central research focus in computer vision and graphics. It has wide-ranging applications in virtual reality, telepresence, and the film industry. These applications are becoming increasingly important as technology advances and the demand for realistic virtual experiences grows. Consequently, a flexible and high-quality 3D human generation system possesses not only immense scientific merit but also practical relevance.

For a high-quality and versatile 3D human generative model, we argue that there are two key factors: **1)** high-capacity generative modeling, and **2)** efficient 3D human representation. Recent advances in 3D-aware generative models [5, 18, 35, 37] have made significant progress. By learning from 2D images using generative adversarial networks (GAN) [14], these methods can synthesize images across different viewpoints in a 3D-aware manner. However, their results still leave significant gaps w.r.t. rendering quality, resolution, and inference speed. Besides, these GAN-based methods often require task-specific adaptations to be conditioned on a particular task, limiting their scope of application. Meanwhile, diffusion-based models [17, 51] have recently surpassed GANs on generative tasks with unprecedented quality, suggesting their great potential in 3D generation. Several recent works [1, 33, 57] have made initial attempts to generate 3D rigid objects based on the diffusion model. Nevertheless, how to integrate the denoising and diffusion processes into a generation framework of articulated 3D humans remains an open problem.

37th Conference on Neural Information Processing Systems (NeurIPS 2023).

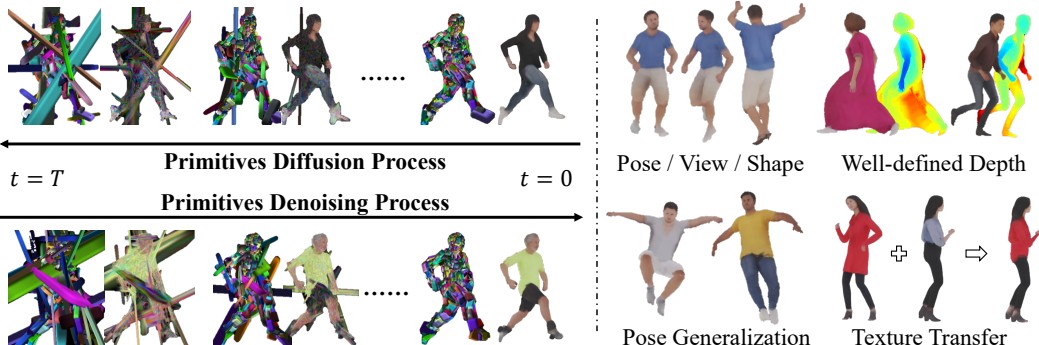

Figure 1: **PrimDiffusion is the first diffusion model for 3D human generation.** (Left) We perform the diffusion and denoising process on a set of primitives which compactly represent 3D humans. (Right) This generative modeling enables explicit pose, view, and shape control, with the capability of modeling off-body topology in well-defined depth. Moreover, our method can generalize to novel poses without post-processing and enable downstream human-centric tasks like 3D texture transfer.

Specifically, applying diffusion models to 3D human generation is a non-trivial task due to the articulated topology, diverse poses, and various identities, among which an efficient human representation is the most important. Unlike human faces [23] and rigid objects [6, 22] where most existing 3D-aware generative models [1, 5, 15, 33] succeed, the human body exists within the vast expanse of 3D space, occupies only a sparse portion of it, characterized by an intricate, articulated topology. An ideal representation should compactly model the human body without wasting parameters on modeling empty space while also enabling explicit view and pose control. The most widely used representation for 3D-aware generative models is based on neural radiance field (NeRF) [32], which can be parameterized by coordinate-based MLP [15, 37] and hybrid representation like tri-plane [5], voxel grids [50] or compositional subnetworks [18]. However, these methods model humans as coarse volumes or planes, which limits their generation quality. In addition, they also suffer from slow inference speeds due to the need for dense MLP decoder queries during rendering.

In this paper, we propose **PrimDiffusion**, the first diffusion model for 3D human generation. Our key insight is that the denoising and diffusion process can be directly operated on a set of volumetric primitives, which models the 3D human body as a number of tiny volumes with radiance and kinematic information. Each volume parameterizes the spatially-varied color and density with six degrees of freedom. This representation fuses the capacity of volumetric representations with the efficiency of primitive-based rendering, which enables: **1)** compact and expressive parameter space for the diffusion model, **2)** flexible representation that inheres human prior, and **3)** efficient and straightforward decoder-free rendering. Furthermore, these primitives naturally capture dense correspondence, facilitating downstream human-centric tasks such as texture transfer and inpainting with 3D consistency. Notably, previous GAN-based methods [18, 61] implicitly encode control signals (*e.g.,* view directions, poses) as inputs, indicating that they need extra forward passes upon condition changes (*e.g.,* novel view/pose synthesis). In contrast, PrimDiffusion supports real-time rendering of high-quality 3D humans at a resolution of $512 \times 512$ once the denoising process is done.

Extensive experiments are conducted both qualitatively and quantitatively, demonstrating the superiority of PrimDiffusion over state-of-the-art methods for 3D human generation. We summarize our contributions as follows: **1)** To the best of our knowledge, we introduce the first diffusion model for 3D human generation. **2)** We propose to represent 3D humans as volumetric primitives in a generative context, which enables efficient training and high-performance rendering. **3)** We design an encoder tailored with cross-modal attention, which accounts for learning volumetric primitives from images across identities without per-subject optimization. **4)** We demonstrate applications of PrimDiffusion, including texture transfer and 3D inpainting, which can be naturally done without retraining.

## 2   Related Work

**3D-aware Generation.**    With recent advances in neural radiance field (NeRF) [32], many studies [5, 9, 11, 15, 18, 35, 37, 49, 50, 59] have incorporated NeRF as the key inductive bias to make GANs be

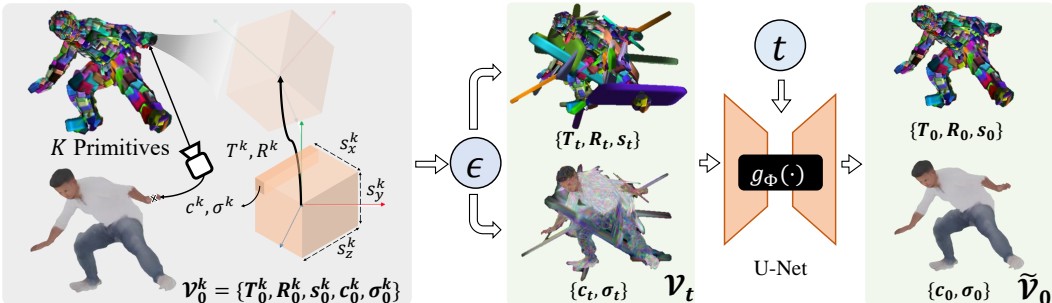

Figure 2: **Overview of PrimDiffusion.** We represent the 3D human as $K$ primitives learned from multi-view images. Each primitive $\mathcal{V}^k$ has independent kinematic parameters $\{\mathbf{T}^k, \mathbf{R}^k, \mathbf{s}^k\}$ (translation, rotation, and per-axis scales, respectively) and radiance parameters $\{\mathbf{c}^k, \sigma^k\}$ (color and density). For each time step $t$, we diffuse the primitives $\mathcal{V}_0$ with noise $\epsilon$ sampled according to a fixed noise schedule. The resulting $\mathcal{V}_t$ is fed to $g_\Phi(\cdot)$ which learns to predict the denoised $\tilde{\mathcal{V}}_0$.

3D-aware, which enables GANs to learn 3D-aware representations from 2D images. Researchers have proposed to use efficient representations like voxel grids [50] and triplanes [5] to improve the spatial resolution of the scene representation, which is a crucial factor for the quality of geometry and images. However, these techniques necessitate an implicit decoder (*e.g.,* a Multilayer Perceptron (MLP) or Convolutional Neural Network (CNN)) following the 3D representation to obtain the rendered images, which results in substantial computational overheads associated with neural volume rendering. Consequently, neither the image quality nor the rendering speed are satisfactory.

**Diffusion Model.** Despite the remarkable success of GAN-based models, diffusion-based models [17, 51, 52] have recently exhibited impressive performance in various generative tasks, especially for 2D tasks like text-to-image synthesis [34, 41, 44, 46]. However, their potential for 3D generation remains largely unexplored. A few attempts have been made on shape [25, 30, 60], point cloud [65], and text-to-3D [39] generation. Recently, some works have succeeded in learning diffusion models from 2D images for unconditional 3D generation [1, 33, 57] of human faces and objects. Still, given the articulated topology of the human body, efficiently training 3D diffusion models is challenging.

**Human Generation.** Extensive efforts have been made to generate human-centric assets [8, 19, 47, 62, 64], especially for 2D human images [12, 13, 21, 24, 48]. For 3D human generation, initial attempts have been made to generate human shapes [7, 56] using scanned data. Several works learn 3D-aware GANs from 2D image collections driven by differentiable rendering. However, they are either in low-resolution [36] or rely on super-resolution module [3, 61] which does not guarantee 3D consistency. EVA3D [18] succeeds in high-resolution 3D human generation within a clean framework. As a concurrent work, HumanGen [20] leverages explicit image priors from strong pretrained 2D human image generators and 3D geometry priors from PIFu [47]. However, these 3D-aware GAN-based methods implicitly condition the viewpoints and poses, which causes additional forward calls when conditions change, preventing high-resolution real-time performance.

## 3 Methodology

The goal of PrimDiffusion is learning to generate 3D humans from multi-view images built upon denoising and diffusion probabilistic models. It learns to revert the diffusion process that gradually corrupts 3D human representations by injecting noise at different scales. In this paper, 3D humans are represented as volumetric primitives, facilitating feasible training of diffusion models and efficient rendering. We first introduce this 3D human representation and corresponding rendering algorithm (Sec. 3.1). Then, a generalizable encoder is employed to robustly fit primitive representations from multi-view images across identities without per-subject optimization (Sec. 3.2). Once the volumetric primitives are learned from images, the problem of 3D human generation is reduced to learning the distribution of 3D representations. We thereby present our denoising and diffusion modeling operated on the parameter space of volumetric primitives to generate high-quality 3D humans (Sec.3.3). An overview of PrimDiffusion is presented in Fig. 2.

### 3.1 Primitive-based 3D Human Representation

To learn a denoising and diffusion model with explicit 3D supervision, we need an expressive and efficient 3D human representation that accounts for **1)** compact parameter space for diffusion models, **2)** articulated topology and motion of 3D humans, and **3)** fast rendering with explicit 3D controls.

**Volumetric Primitives.**    Inspired by previous work [28, 43] in reconstructing dynamic human heads and actors, we propose to represent 3D humans as a set of volumetric primitives which acts as the target space for diffusion models. Note that, whereas previous research has centered on the per-subject reconstruction, we enable learning volumetric primitives across multiple identities. This paves the way for a shared latent space conducive to generative modeling. In specific, this representation consists of a set of $K$ primitives that jointly parameterize the spatially varied color and density:

$$\mathcal{V} = \{\mathcal{V}^k\}_{k=1}^K, \quad \text{where} \quad \mathcal{V}^k = \{\mathbf{T}^k, \mathbf{R}^k, \mathbf{s}^k, \mathbf{c}^k, \sigma^k\}. \tag{1}$$

Each primitive $\mathcal{V}^k$ is a tiny volume, parameterized by a translation vector $\mathbf{T}^k \in \mathbb{R}^3$, an orientation $\mathbf{R}^k \in SO(3)$, a three-dimensional scale factor $\mathbf{s}^k \in \mathbb{R}^3$, a spatially-varied color $\mathbf{c}^k \in \mathbb{R}^{3 \times S \times S \times S}$, and density $\sigma^k \in \mathbb{R}^{S \times S \times S}$, where $S$ is the per-axis base resolution. In practice, we set the base resolution of one primitive to $S = 8$, which we find the best to strike a balance between reconstruction quality and memory consumption.

To model humans with articulated topology and motion, we attach primitives to the vertices of SMPL [29]. SMPL is a parametric human model, defined as $\mathcal{M}(\beta, \theta) \in \mathbb{R}^{6890 \times 3}$, where $\beta \in \mathbb{R}^{10}$ controls body shapes and $\theta \in \mathbb{R}^{72}$ controls body poses. In this paper, we leverage this parametric human model for the purposes of 1) serving as a geometry proxy to weakly constrain volumetric primitive representations, 2) offering geometry cues for 3D representation fitting, and 3) generalizing to novel poses through linear blending skinning (LBS) algorithm tailored with decoder-free rendering.

Specifically, the kinematic parameters of primitives are modeled relative to the base transformation from SMPL. Namely, we generate a $W \times W$ 2D grid in the UV space of SMPL. The 3D positions of primitives are initialized by uniformly sampling UV space and mapping each primitive to the closest surface point on SMPL that corresponds to the UV coordinates of the grid points. The orientation of each primitive is set to the local tangent frame of the 3D surface point. The per-axis scale is calculated based on the gradient of the UV coordinates at the corresponding grid point. All primitives are given an initial scale that is proportional to the distances between them and their neighbors. Furthermore, to model off-body topologies like loose clothes and fluffy hair, the scale of primitives is allowed to deviate from the base scale $\hat{\mathbf{s}}_k$. The scale factor is defined as $\mathbf{s}_k = \hat{\mathbf{s}}_k \cdot \delta \mathbf{s}_k$, where $\hat{\mathbf{s}}_k$ is the aforementioned initialized base scale and $\delta \mathbf{s}_k$ is the delta scaling prediction.

Therefore, the learnable parameters of each person modeled by its volumetric primitives $\mathcal{V}$ are color $\mathbf{c} \in \mathbb{R}^{W \times W \times 3 \times S \times S \times S}$, density $\sigma \in \mathbb{R}^{W \times W \times 1 \times S \times S \times S}$, and delta scale factor $\delta \mathbf{s} \in \mathbb{R}^{W \times W \times 3}$.

**Efficient Decoder-free Rendering.**    Once the radiance and kinematic information of primitives are determined, we can leverage differentiable ray marching [27] to render the corresponding camera view. For a given camera ray $\mathbf{r}_p(t) = \mathbf{o}_p + t\mathbf{d}_p$ with origin $\mathbf{o}_p \in \mathbb{R}^3$ and ray direction $\mathbf{d}_p \in \mathbb{R}^3$, the corresponding pixel value $I_p$ is calculated as an integral:

$$I_p = \int_{t_{\min}}^{t_{\max}} \mathbf{c}(\mathbf{r}_p(t)) \cdot \frac{dT(t)}{dt} \cdot dt, \quad \text{where} \quad T(t) = \int_{t_{\min}}^{t} \sigma(\mathbf{r}_p(t)) \cdot dt, \tag{2}$$

where $t_{\min}$ and $t_{\max}$ are the near and far bound for points sampling. The $\mathbf{c}(\cdot)$ and $\sigma(\cdot)$ are global color and density fields derived from trilinear interpolation of color and density attributes among primitives hit by the camera ray. Notably, in contrast to NeRF-based methods [32] which often require evaluating MLP decoders per camera rays, this rendering process is decoder-free which can be directly rendered into pixels. Thus, it is far more efficient for both training and rendering.

To animate the volumetric primitives given a pose sequence $\{\theta_j\}_{j=1}^{N_{\text{frame}}}$, we explicitly employ the LBS algorithm on top of SMPL [29] to produce per-frame human mesh $\mathcal{M}(\beta, \theta_j)$, which serve as driving signal to determine the kinematic information of the moving volumetric primitives.

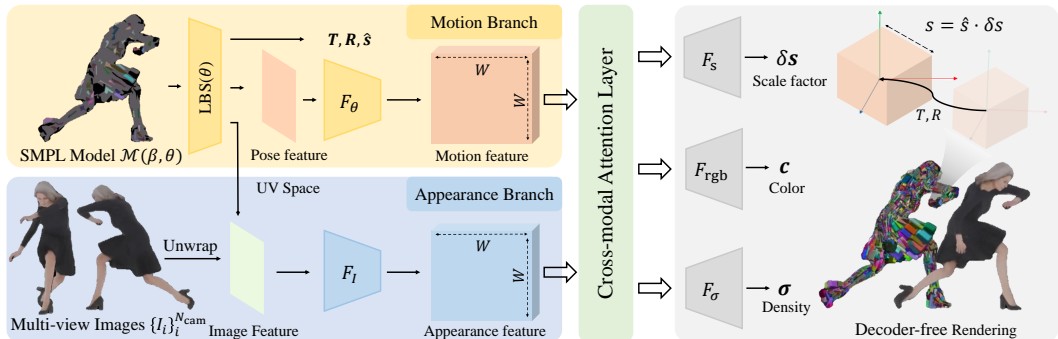

Figure 3: **Encoder architecture for generalizable primitives fitting.** To get rid of per-subject optimization, we propose an encoder that is capable of learning primitives from multi-view images across identities. The encoder consists of a motion branch and an appearance branch, which are fused by the proposed cross-modal attention layer to get kinematic and radiance information of primitives.

## 3.2 Generalizable Primitive Learning from Multi-View Images

In order to leverage explicit 3D supervision from multi-view images, it is essential to reconstruct 3D representations for each individual to provide ground truth for the diffusion model. While earlier research [57] on 3D diffusion models adhered to a per-identity fitting fashion, which could be time-consuming on a large-scale dataset. To overcome this limitation, we propose a generalizable encoder to reconstruct 3D human representation across diverse identities without per-subject optimization.

Our setting is learning 3D representations from multi-view images $\{\mathcal{I}_i\}_{i=1}^{N_{cam}}$, where $N_{cam}$ denotes the number of camera views and $i$ is the corresponding index. We first estimate the SMPL [29] parameters $\mathcal{M}(\beta, \theta)$ using off-the-shelf tools [10]. Then, we employ an encoder with dual branches to parameterize the coarse human mesh as well as the multi-view images, which outputs the radiance and kinematic information of volumetric primitives. The architecture is illustrated in Figure 3.

Specifically, the encoder consists of two branches, one is for appearance while the other is for motion. For the appearance branch, we back-project the pixels of input images corresponding to all visible vertices of the SMPL model to the UV space. The unwrapped image textures will be averaged across multiple views and fed to the image encoder $F_I(\mathcal{I}_i; \Phi_I)$ parameterized by $\Phi_I$, providing texel-aligned appearance features. For the motion branch, we take as input the local pose parameters of SMPL model, *i.e.,* $\theta$ excluding the first three elements that denote global orientation. The local pose parameters are padded to a $W \times W$ 2D map and fed to the motion encoder $F_\theta(\theta; \Phi_\theta)$ parameterized by $\Phi_\theta$, providing motion features. Instead of fusing the features from two branches via naive concatenation, we propose a cross-modal attention module to effectively capture the dependencies between appearance and motion, which significantly improves the reconstruction quality of 3D humans (Tab. 3 and Fig. 7). Intuitively, this module captures the interdependence of color and motion, *e.g.,* clothes wrinkles and shadows caused by different human poses. Then, the radiance and kinematic information (color, density, and delta scale) of volumetric primitives are separately predicted by three mapping networks $(F_{rgb}, F_\sigma, F_s)$ conditioned on the fused feature.

The encoder comprises $\{F_I, F_\theta, F_{rgb}, F_\sigma, F_s\}$ is trained in an end-to-end manner across identities instead of per-subject optimization. The learning objective is formulated as a reconstruction task between ground truth images and rendered images:

$$\mathcal{L}_{rec} = \lambda_{rgb}\mathcal{L}_{rgb} + \lambda_{sil}\mathcal{L}_{sil} + \lambda_{vol}\mathcal{L}_{vol}, \tag{3}$$

where $\mathcal{L}_{rgb}$ is the image reconstruction loss in L2-norm, $\mathcal{L}_{sil}$ is the silhouette loss, $\mathcal{L}_{vol}$ is the volume regularization term [28], and $\lambda_*$ are loss weights. The volume regularization is defined as $\mathcal{L}_{vol} = \sum_{i=1}^{K} \text{Prod}(s_i)$, where $\text{Prod}(\cdot)$ denotes the product of scale factor along three axes. It aims to prevent large primitives from occupying empty space that leads to loss of resolution.

## 3.3 Primitive Diffusion Model

Once the volumetric primitives are reconstructed, the problem of 3D human generation is reduced to learning the distribution $p(\mathcal{V})$ with diffusion models. In specific, to generate a 3D human with its

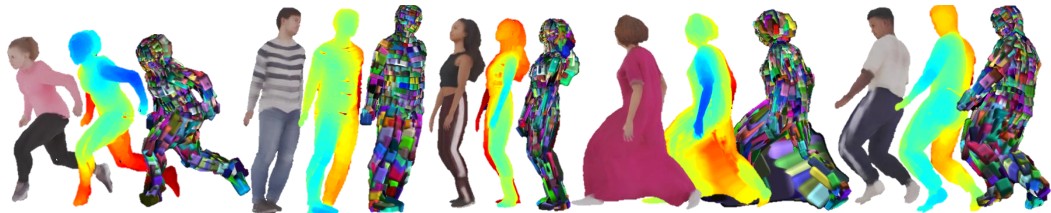

Figure 4: **Generated 3D humans of PrimDiffusion.** Our method can synthesize 3D humans with explicit controls of view, pose and shape, together with well-defined depths. Additionally, we can also handle off-body topologies such as bloomy hair and loose garments using primitives. We visualize the RGB, depth, and primitives side-by-side. Please check the supplementary video for more results.

representation as $\mathcal{V}_0$, the diffusion model learns to denoise $\mathcal{V}_T \sim \mathcal{N}(\mathbf{0}, \mathbf{I})$ progressively closer to the data distribution through denoising steps $\{\mathcal{V}_{T-1}, \dots, \mathcal{V}_0\}$. Note that, the targets of denoising and diffusion process are color $\mathbf{c} \in \mathbb{R}^{W \times W \times 3 \times S \times S \times S}$, density $\sigma \in \mathbb{R}^{W \times W \times 1 \times S \times S \times S}$, and delta scale factor $\delta\mathbf{s} \in \mathbb{R}^{W \times W \times 3}$. To form a parameter space with a regular shape, we first pad the delta scale $\delta\mathbf{s}$ to the same size of color $\mathbf{c}$. Then, we concatenate and reshape them into a 3D tensor. With slight abuse of notation, we denote this tensor as $\mathcal{V}_0 \in \mathbb{R}^{[W \cdot S] \times [W \cdot S] \times [7 \cdot S]}$ in the following subsections.

**Forward Process.** Given a data point, *i.e.,* volumetric primitives of a human, sampled from a real data distribution $\mathcal{V}_0 \sim q(\mathcal{V})$, we define the forward diffusion process in which we add Gaussian noise to the sample in $T$ steps, producing a sequence of noisy samples $\{\mathcal{V}_1, \dots, \mathcal{V}_T\}$:

$$q(\mathcal{V}_t | \mathcal{V}_{t-1}) = \mathcal{N}(\mathcal{V}_t; \sqrt{1-\beta_t}\mathcal{V}_{t-1}, \beta_t \mathbf{I}), \tag{4}$$

where the step sizes are controlled by a variance schedule $\{\beta_t \in (0, 1)\}_{t=1}^{T}$. We use a linear scheduler [17] in our experiments. Denoting $\alpha_t = 1 - \beta_t$ and $\bar{\alpha}_t = \prod_{i=1}^{t} \alpha_i$, we can directly sample $\mathcal{V}_t$ at any arbitrary time step $t$ in a closed form through reparameterization:

$$q(\mathcal{V}_t | \mathcal{V}_0) = \prod_{i=1}^{t} q(\mathcal{V}_i | \mathcal{V}_{i-1}) = \mathcal{N}(\mathcal{V}_t; \sqrt{\bar{\alpha}_t}\mathcal{V}_0, (1 - \bar{\alpha}_t)\mathbf{I}). \tag{5}$$

**Reverse Process.** The reverse process, namely the denoising process, aims to generate a true sample $\mathcal{V}_0$ from a Gaussian noise $\mathcal{V}_T \sim \mathcal{N}(\mathbf{0}, \mathbf{I})$, which requires an estimation of the posterior distribution:

$$q(\mathcal{V}_{t-1} | \mathcal{V}_t, \mathcal{V}_0) = \mathcal{N}(\mathcal{V}_{t-1}; \tilde{\mu}(\mathcal{V}_t, \mathcal{V}_0), \tilde{\beta}_t \mathbf{I}), \quad \tilde{\mu}(\mathcal{V}_t, \mathcal{V}_0) = \frac{\sqrt{\alpha_t}(1 - \bar{\alpha}_{t-1})}{1 - \bar{\alpha}_t}\mathcal{V}_t + \frac{\sqrt{\bar{\alpha}_{t-1}}\beta_t}{1 - \bar{\alpha}_t}\mathcal{V}_0, \tag{6}$$

where $\tilde{\beta}_t = (1 - \bar{\alpha}_{t-1})/(1 - \bar{\alpha}_t)\beta_t$. Notably, $\mathcal{V}_0$ is the generation target that is unknown. Thus, we need to learn a denoiser $g_\Phi$ parameterized by $\Phi$ to approximate these conditional probabilities $p_\Phi(\mathcal{V}_{t-1} | \mathcal{V}_t) \approx q(\mathcal{V}_{t-1} | \mathcal{V}_t, \mathcal{V}_0)$, *i.e.,* estimate the mean $\tilde{\mu}(\cdot)$. In practice, we train the denoiser $g_\Phi$ to predict $\mathcal{V}_0$ such that:

$$\mu_\Phi(\mathcal{V}_t, t) = \frac{1}{\sqrt{\alpha_t}}\left(\mathcal{V}_t - \frac{1-\alpha_t}{1-\bar{\alpha}_t}(\mathcal{V}_t - \sqrt{\bar{\alpha}_t}g_\Phi(\mathcal{V}_t, t))\right), \tag{7}$$

where the posterior probabilities can be approximated as $p_\Phi(\mathcal{V}_{t-1} | \mathcal{V}_t) = \mathcal{N}(\mathcal{V}_{t-1}; \mu_\Phi(\mathcal{V}_t, t), \tilde{\beta}_t \mathbf{I})$. During inference, this posterior is sampled at each time step $t$ to gradually denoise the noisy sample $\mathcal{V}_t$. The denoiser $g_\Phi$ is trained to minimize the difference from $\tilde{\mu}$ which can be simplified [17] as :

$$\mathcal{L}_t^{\text{simple}} = \mathbb{E}_{t \sim [1, T], \mathcal{V}_0, \mathcal{V}_t}[||\mathcal{V}_0 - g_\Phi(\mathcal{V}_t, t)||_2^2]. \tag{8}$$

## 4 Experiments

**Dataset.** We obtain 796 high-quality 3D humans from RenderPeople [55] with diverse identities and clothes. For each person, we repose the mesh with 20 different human poses [31] to ensure pose diversity. For each pose instance of a person, we render 36 multi-view images with known camera poses. All methods are trained from scratch (except mentioned) on this dataset for fair comparisons.

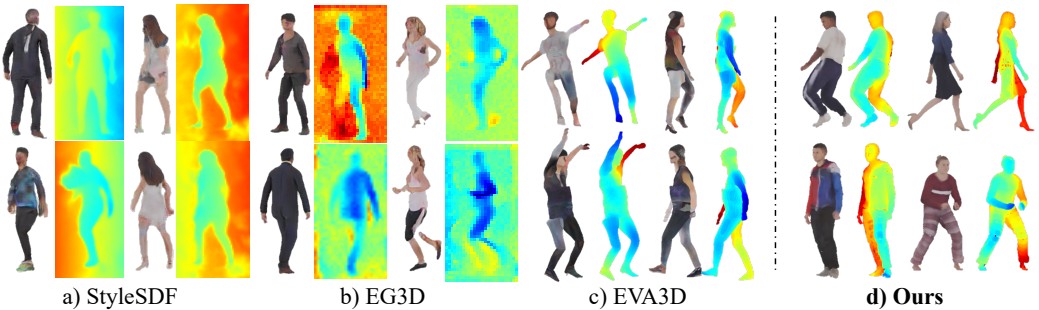

| a) StyleSDF | b) EG3D | c) EVA3D | **d) Ours** |

Figure 5: **Qualitative comparisons of unconditional 3D human generation between PrimDiffusion and baseline methods.** RGB renderings and corresponding depths are placed side-by-side.

Table 1: **Quantitative results on unconditional 3D human generation.** The top three techniques are highlighted in red, orange, and yellow, respectively. In this table, "ft" indicates a finetuned model that is pre-trained on the DeepFashion [26] dataset.

| Methods | $\text{FID}_{\text{CLIP}} \downarrow$ | FID $\downarrow$ | KID$\times 10^2 \downarrow$ | PCK $\uparrow$ | Depth$\times 10^2 \downarrow$ | FPS $\uparrow$ |
|---|---|---|---|---|---|---|
| StyleSDF [37] | 18.55 | 51.27 | $4.08 \pm 0.13$ | - | $49.37 \pm 22.18$ | 2.53 |
| EG3D [5] | 19.54 | 24.32 | $1.96 \pm 0.10$ | - | $16.59 \pm 21.03$ | 22.97 |
| EVA3D [18] | 15.03 | 44.37 | $2.68 \pm 0.13$ | 91.84 | $3.24 \pm 9.93$ | 6.14 |
| EVA3D ft [18] | 14.58 | 40.40 | $2.99 \pm 0.14$ | 91.30 | $2.60 \pm 7.95$ | 6.14 |
| **Ours** | 12.11 | 17.95 | $1.63 \pm 0.09$ | 97.62 | $1.42 \pm 1.78$ | 88.24 |

**Implementation Details.** We use $K = W^2 = 1024$ primitives to represent each 3D human. The denoiser $g_\Phi$ is implemented as a 2D U-Net [45] with intermediate attention layers. We first train the generalizable encoder using all available images. Then, the denoiser $g_\Phi$ is trained according to the primitives produced by the frozen encoder. Please check the supplementary material for more details.

**Evaluation Metrics.** We adopt Fréchet Inception Distance (FID) [16] and Kernel Inception Distance (KID) [4] to evaluate the quality of rendered images. Note that, we use different backbones to evaluate FID in different latent spaces in order to get a thorough evaluation, *i.e.,* $\text{FID}_{\text{CLIP}}$ employs the CLIP image encoder [40] while FID employs the Inception-V3 model [54]. To evaluate the 3D geometry, we use an off-the-shelf tool [42] to estimate depth maps from renderings and compute L2 distance against rendered depths. We further adopt a human-centric metric, Percentage of Correct Keypoints (PCK) [2], to evaluate the pose controllability of 3D human generative models.

**Comparison Methods.** As the first diffusion model for 3D human generation, we compare with three GAN-based methods. EVA3D [18] learns to generate 3D human from 2D image collections. EG3D [5] and StyleSDF [37] are approaches for 3D-aware generation from 2D images, succeeding in generating human faces and objects. All of these methods enable view control by implicitly conditioning on the camera pose, which leads to extra forward passes upon viewpoint changes.

### 4.1 Qualitative Results

We show the RGB renderings and corresponding depth maps generated by PrimDiffusion and baselines in Fig. 5. StyleSDF and EG3D achieve coarse results in terms of appearance and geometry. However, due to the lack of human prior, most of the capacity is wasted on modeling the empty space that is not occupied by the human body, which leads to blurry facial details. Furthermore, the inefficient 3D representations make them hard to scale up to high resolution without super-resolution modules, which also limits their geometry to low resolution ($64^2$ or $128^2$). EVA3D succeeds in high-resolution rendering with much better geometry by incorporating human prior. However, it fails to generate enough facial details. We attribute this to its part-wise human representation. For example, EVA3D models the human head as a single volume, assigning a difficult learning problem of distinguishing facial details within one uniform representation. In contrast, PrimDiffusion divides humans into many primitives tailed with powerful diffusion modeling, which is not only capable of

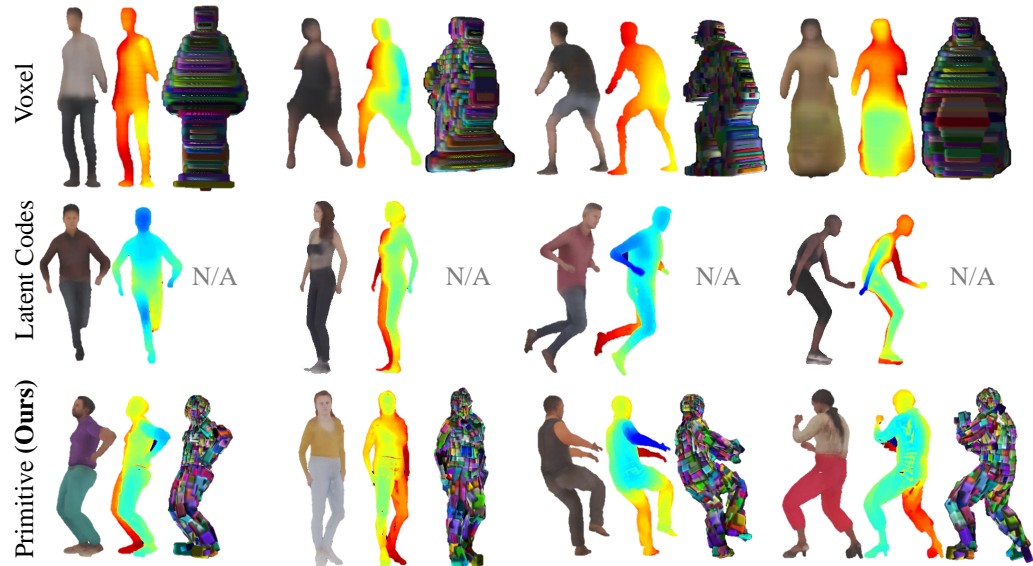

Figure 6: **Qualitative ablation study of different human representations for 3D diffusion.** Alternatively, we replace primitives with voxel grids as human representation to validate the effectiveness. RGB renderings, depths, and representation occupancy are placed sequentially for each sample.

Table 2: **Quantitative ablation study of different human representations for 3D diffusion.** Note that, the "Resolution" denotes the 3D spatial resolution of the representation.

| Methods | Resolution | $\text{FID}_{\text{CLIP}} \downarrow$ | FID $\downarrow$ | KID$\times 10^2 \downarrow$ | Depth$\times 10^2 \downarrow$ |
|---|---|---|---|---|---|
| Voxel | $64 \times 64 \times 64$ | 19.53 | 81.78 | $7.72 \pm 0.17$ | $8.22 \pm 3.78$ |
| Latents [38] | - | 12.13 | 37.70 | $3.44 \pm 0.11$ | $1.76 \pm 1.91$ |
| **Ours** | $256 \times 256 \times 8$ | 12.11 | 17.95 | $1.63 \pm 0.09$ | $1.42 \pm 1.78$ |

modeling enough details (*e.g.,* faces and textures) but also supporting $512 \times 512$ resolution for both RGB and depth without any super-resolution decoder.

## 4.2 Quantitative Results

The results of numerical comparisons are presented in Tab. 1. As general 3D generation approaches, StyleSDF and EG3D get fair scores in terms of generation quality, but the geometries are much worse than others. EVA3D generates good geometries, indicated by its comparable depth error with us. However, the image quality is still far behind us. Note that, we observe the vanilla FID might be inaccurate, which leads to discrepancies between visualizations and numerical results. For example, the images rendered by EG3D are of poor quality but the FID is much lower than other baselines. Therefore, we report $\text{FID}_{\text{CLIP}}$ metric which we found more consistent with qualitative results. Nevertheless, the proposed method still outperforms baselines by a large margin.

Moreover, we also compare our model with a finetuned EVA3D that is first pre-trained on DeepFashion [26] dataset to establish 3D representations of humans and then finetuned on our RenderPeople dataset. The motivation for this experiment is to fully leverage the ability to learn 3D representations on both multiview images and collections of images for methods like EVA3D instead of training from scratch with multiview images only. However, as shown in Tab. 1, the finetuned EVA3D still has a performance gap compared with our method.

## 4.3 Ablation Studies

**Different Representations for 3D Diffusion.** Apparently, the volumetric primitive is not the only representation for 3D human diffusion models. We also explore the effectiveness of other representations, like voxel grids [53] and latent codes [38]. The "voxel" baseline differentiates itself

Table 3: **Ablation study of design choices for volumetric primitives fitting.** The metrics PSNR, SSIM [58], and LPIPS [63] are averaged across all training identities, views, and poses.

| Methods | PSNR ↑ | SSIM ↑ | LPIPS ↓ |
|---|---|---|---|
| w/o varying $\delta s$ | 18.35 | 0.803 | 0.182 |
| w/o $R, T$ | 31.70 | 0.982 | 0.058 |
| w/o attention | 28.39 | 0.962 | 0.081 |
| **Ours** | **32.15** | **0.984** | **0.048** |

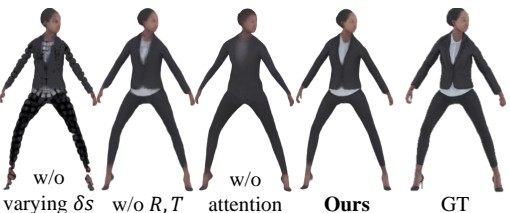

w/o varying $\delta s$    w/o $R, T$    w/o attention    **Ours**    GT

Figure 7: **Visualizations on effects of different design choices for volumetric primitives fitting.** "GT" denotes the ground truth image.

from our method in two ways: 1) it utilizes voxel grids with a resolution of $64 \times 64 \times 64$ for fitting 3D representation from multi-view images, and 2) it leverages 3D convolution layers instead of 2D for instantiation of $g_\Phi(\cdot)$. Note that, as 3D convolution layers occupy much more memories than 2D, the feasible resolution of voxel grids is rather limited. Moreover, the "latents" baseline inherits the idea of anchoring latent codes [38] to the vertices of the SMPL model for 3D human representation learning. However, the vanilla NeuralBody [38] takes 14 hours per subject to converge which makes it unscalable to large-scale datasets like RenderPeople which contains lots of identities. Thus, we implement the "latents" baseline in the following way. We replace the payload of volumetric primitives with latent codes and leverage sparse convolution layers as the decoder to output the color and density of radiance fields. This implementation of a NeuralBody version of our method keeps the ability to learn cross-identity 3D representation through our generalizable primitive learning framework while getting rid of per-subject fitting.

Quantitative results are presented in Tab. 2. The voxel-based representation fails to achieve a reasonable result both in rendering quality and geometry. The latent codes baseline achieves plausible generated results but worse quality compared with ours. We further visualize the qualitative comparisons in Fig. 6. The voxel-based approach wastes many parameters to densely model the scene, leading to the low spatial resolution of the representation. This is the root cause of its blurry renderings, which validates the importance of our primitive-based representation for 3D human diffusion model.

**Ablation on Design Choices for Primitives Fitting.** The quality of primitives fitting significantly affects the generation quality, as it provides 3D ground truth for the diffusion model. Therefore, we ablate the effectiveness of different designs for generalizable primitives fitting, presented in Fig 7 and Tab. 3. Note that, we use reconstruction metrics for evaluation. We denote "w/o varying $\delta s$" as the baseline that replaces the spatially-varied delta scale factor $\delta s$ with a uniformly distributed scale factor, *i.e.,* all primitives share the same delta scale factor. It fails to reconstruct reasonable 3D humans. The "w/o $R, T$" denotes the method that uses predefined rotation and translation calculated from a template human mesh instead of an identity-specific one, which leads to a slight drop in quality. And "w/o attention" denotes the method trained without our proposed cross-modal attention module, which is important for detailed textures and motion-dependent effects.

## 4.4 Further Analysis

**Real-time Inference.** Thanks to our efficient representation and decoder-free rendering procedure, PrimDiffusion can render 3D consistent humans with varying viewpoints and poses in a resolution of $512 \times 512$ at 88.24 FPS once the denoising process is done. As shown in Tab. 1, due to the use of the decoder which decodes the 3D representation to images, all baselines fail to high-resolution real-time rendering at a comparable quality with us. Furthermore, these methods implicitly model 3D-aware conditions, *i.e.,* viewpoints and poses, which causes extra forward pass calls upon conditions changes.

**Generalizability to Novel Poses.** It is worth mentioning that a decoder (*e.g.,* MLP for NeRF decoding or CNN for super-resolution) may lead to overfitting to the pose distribution of the training dataset, which prohibits pose generalization to out-of-distribution. Another merit of our decoder-free rendering is that we can generalize to novel poses without test-time optimization, post-processing, and rigging (Fig. 1 and Fig. 4). We refer readers to the supplementary material for video results.

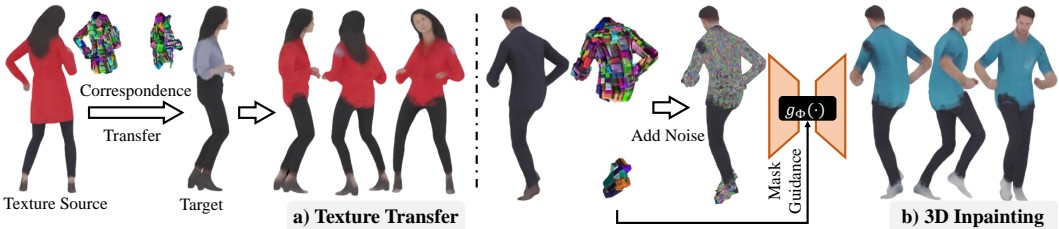

Figure 8: **Application of PrimDiffusion.** a) Thanks to the dense correspondence offered by volumetric primitives, we can transfer the texture in a 3D consistent way. b) By masking the corresponding primitives with Gaussian noise and performing the denoising process with mask guidance, our method can achieve the conditional generative task, *i.e.,* 3D inpainting, without retraining the model.

### 4.5 Applications

**Texture Transfer.** Based on the dense correspondence offered by volumetric primitives, we can transfer the texture from one human body to another, as shown in Fig. 8 a). Since the entire process is operated in 3D space, we can render the human with transferred texture with free viewpoints.

**3D Inpainting.** Thanks to the flexibility of primitive-based representation and diffusion model, PrimDiffusion can inpaint the masked region on the human body with new textures as shown in Fig. 8 b). By perturbing the masked region with noise, we take the volumetric primitives as the mask guidance to the denoiser $g_\Phi(\cdot)$. Note that, in contrast to GAN-based methods, this conditional generation is done without retraining the model.

## 5   Conclusion

In this paper, we propose the first diffusion model for 3D human generation. To offer a compact and expressive parameter space with human prior for diffusion models, we propose to directly operate the diffusion and denoising process on a set of volumetric primitives, which models the human body as a number of small volumes with radiance and kinematic information. This flexible yet efficient representation enables high-performance rendering for novel views and poses, with a resolution of $512 \times 512$ in real-time. Furthermore, we also propose a cross-modal attention module tailored with an encoder-based pipeline for fitting volumetric primitives from multi-view images without per-subject optimization. We believe our method will pave the way for further exploration in 3D human generation.

**Limitation.** While our method shows promising results for unconditional 3D human generation, several limitations remain. First, though we can model off-body topology like loose garments and long hair, how to animate them remains a challenge. Second, our renderings contain artifacts for complex textures. Leveraging 2D pre-trained generative models would be a possible solution. Last, due to the high degree of freedom of primitives, our method is a two-staged pipeline to prevent training instability. Further explorations on single-stage 3D diffusion models would be fruitful. In addition, compared with GAN-based approaches, our method requires a sufficient number of posed views to get good 3D representations for training diffusion models. In this context, it would be interesting to explore few-shot inverse rendering techniques to reduce the reliance on multiview images for the first stage of our method.

**Broader Impacts.** The generated 3D humans of PrimDiffusion might be misused to create misleading content or fake media. It can also edit personal imagery, leading to privacy concerns.

## Acknowledgments and Disclosure of Funding

This work is supported by the National Research Foundation, Singapore under its AI Singapore Programme (AISG Award No: AISG2-PhD-2021-08-019), NTU NAP, MOE AcRF Tier 2 (T2EP20221-0012), and under the RIE2020 Industry Alignment Fund - Industry Collaboration Projects (IAF-ICP) Funding Initiative, as well as cash and in-kind contribution from the industry partner(s).

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
