# OpenReview forum: "PrimDiffusion: Volumetric Primitives Diffusion for 3D Human Generation"
_NeurIPS.cc/2023/Conference — NeurIPS 2023 poster_

### Official Review · Reviewer_GMsy · 2023-06-13

**Soundness:** 2 fair
**Presentation:** 3 good
**Contribution:** 2 fair
**Rating:** 4
**Confidence:** 4

**Summary:**

This paper introduces a method to generate various 3D human geometry and appearance. To this end, it introduces a new design of diffusion model that learns to generate the parameters of 3D primitives of humans: a generalizable encoder constructs motion and appearance features and their attentions are fed to predict scale, color, and density of each primitive. The evaluation is performed on the render people dataset, and the authors demonstrate a number of applications such as reposing and 3D inpainting.

**Strengths:**

+ This paper effectively combines existing 3D diffusion model and existing 3D representation to perform generalizable 3D human generation. It is the first attempt to combine diffusion model and primitive representation for 3D generative humans. It can serve as good initial baseline for future research.
+ The 3D inpainting application is interesting, which allows reconstructing a complete 3D human from a partial body.


**Weaknesses:**

1] The motivation of this paper is not clear.
In particular, it is not clear why the volume primitive representation is ideal (other than efficiency) for handling complex human articulations. Other than computational efficiency, why does this representation fundamentally enables better quality of 3D human generation than other representation (e.g., point clouds, sdf, occupancy, and so on)?
Simply showing the ablation study (without thorough analysis) does not account for this fundamental motivation, and I would like to request the authors to explain above question with fundamental reasons. Without strong motivation, this work will fall into the work that just simply combines existing 3D diffusion with existing representation as one try of a baseline.

2] Misleading claims and overclaims:
2-1) In L45, existing methods (nerf or tri-plane) do not model humans as coarse volumes. They model the geometry in the continuous space, which can lead to the fine-results. In fact, the coarseness is quite proportional to the resolution (e.g., ray resolution, volume resolution). In that sense, the proposed method could be considered as coarse reconstruction method (also, the geometry and texture quality seem coarse).
2-2) Claiming the proposed method as real-time approach sounds overstated since the generation time of diffusion model is quite slow. Please claim the computational performance in light of the whole pipeline.


3] Weak clarity in implementation.
3-1) It is not clear how the authors repose the human from the generated 3D volume primitives while preserving identity.
3-2) While the number of parameters are five, the authors describe it as six-degree of freedom. Why six-degree of freedom?
3-3) Figure 2 is not clear. The {input, output} modality of Unet should be 2D image space. However, the current modality is not image but the parameters of {T,R,s,c,sigma}. How did you handle the modality for such parameter prediction?
3-4) In L165, it is not clear how the method can handle the texture outside the SMPL body parts.

4] Low-quality results.
4-1) The quality of the generated 3D human geometry and texture is low. In particular, the high-frequency details are highly lacking for both geometry and texture.
4-2) Also, the texture is not reflective of geometry. For example, in Figure 1, the shade in the red dress of the women (of the upper right corner) does not reflect the depth information at all.
4-3) Compared to existing work [1], the 3D generation quality is low.
Are they all mainly due to the weakness of the 3D primitive representation?


5] Weak validation, comparison, demonstration, and analysis:
5-1) missing baseline for 3D human generation. In particular, this paper “must” be compared to gDNA [1] paper, which can generate 3D human geometry conditioned by SMPL model whose objective is exactly same as the one in this paer. Without this comparison, this paper should not claim “state-of-the-art” method;
5-2) Since this paper claims good “computational efficiency”, it should provide the measure of inference time including denoising process, not only the rendering part;
5-3) Due to the use of synthetic data, the human generation results look always synthetic and fake. It would be nice if the authors provide the results of the model trained on real multiview data, e.g., Thuman data [2].
5-4) In L263 and Figure 6, it is unexpected that the voxel-based approach fails that much. Please provide a thorough insight and analysis why the voxel-based approach fails.


[1] gDNA: Towards Generative Detailed Neural Avatars
[2] Function4D: Real-time Human Volumetric Capture from Very Sparse RGBD Sensors







**Questions:**

Please address the questions in the weakness section.

**Limitations:**

This paper sufficiently discussed the limitations.

---

> ### Author Rebuttal · Authors · 2023-08-09
>
> Thank you for your careful review and constructive suggestions. We summarize and answer your questions as follows:
>
> **Q1: Fundamental motivation of the proposed method.**
>
> Please kindly refer to the above post named "Response to the common issue of the novelty and motivation".
>
> **Q2-1: Clarification for the coarseness.**
>
> Thanks for your question. We argue that the coarseness of 3D representations indicates the ratio of effective parameters in modeling the human body. As pointed out in L40, the human body exists within the vast expanse of 3D space, occupies only a sparse portion of it. An ideal representation should compactly model the human body without wasting parameters on modeling empty space. We present the intuition behind in the experiment below. As the accurate volume of the human body is hard to measure, we define the "Efficiency" of a certain 3D representation by projecting them onto camera views. For example, given a rendered human image, we render the occupied space of a 3D representation by setting the volume density to 1. Then we calculate the ratio between the foreground of the human and the projected 3D representation. This efficiency is calculated over 200 views where the camera orbits around the target human body. We present the result below, where the volumetric primitive has the highest efficiency, indicating it is the finest one to represent 3D humans given the computational budget.
>
> |Methods|Efficiency(\%)|FPS|
> |:-:|:-:|:-:|
> |StyleSDF (SDF)|12.02$\pm$1.96|2.53|
> |EG3D (Tri-plane)|12.02$\pm$1.96|22.97|
> |EVA3D (Compositional voxels)|75.27$\pm$1.51|6.14|
> |Ours (Volumetric primitives)|87.24$\pm$1.25|88.24|
>
> **Q2-2: The computational performance in light of the whole pipeline.**
>
> Please refer to the above post named "Response to the common issue of the computational cost".
>
> **Q3: Clarifications of implementation.**
>
> Let us elaborate on details case by case as follows.
>
> 3-1) We repose the human by explicitly reposing the SMPL model through linear blending skinning. Our key insight is that the decoder-free rendering of volumetric primitives enables all primitives to learn the color and density and kinematic parameters in a physically correct way instead of an implicit way. Therefore, reposing only changes the kinematic parameter of primitives while keeping the identity unchanged.
>
> 3-2) The actual degree of freedom is 13 since $T,R,s,c$ are three-dimensional vectors. The description of six degrees of freedom aims to highlight that each primitive has independent rotation and translation which differentiate from prior work where the voxel cannot rotate freely. Will polish L116 in our revised version.
>
> 3-3) As you correctly pointed out, the input and output of Unet should be in 2D image space. Therefore, it is not straightforward to apply diffusion models to volumetric primitives. As we iterated in L140, the learnable parameters contain color $c$, density $s$, and delta scale factor $\delta s$. Then, we clearly provide the detail in L190-L192 on how to form a parameter space of regular shape as the input of Unet. Specifically, we first pad the delta scale $\delta s$ to the same size of color $c$, and concatenate them together into a tensor as $\mathcal{V}_0 \in R^{[W\cdot S]\times[W\cdot S]\times [7S]}$.
>
> 3-4) In L165, we describe how to obtain image-based input feature in the UV space for the appearance branch. Note that, there is no need for this input feature to align with off-body topology like loose garments. We handle loose garments by the independent delta scale factor of each primitive, which is optimized through rendering reconstruction loss.
>
> **Q4: Quality of results.**
>
> Thanks for your comments. Indeed, modeling shadows that are consistent with 3D geometry is an important and interesting future work for 3D human generative models. It requires the underlying 3D human representation to be as efficient as possible for simulating complex light transport, where the volumetric primitive has a chance for its efficiency. Moreover, we demonstrate an impressive ability to model off-body topology (shown as the red dress of the women in Figure 1), which is impossible for prior works like EG3D and EVA3D.
>
> **Q5: Comparisons with gDNA.**
>
> Thanks for your reference. To the best of our knowledge, gDNA is an awesome model for generating a large variety of animatable 3D human shapes. However, our task is to generate textured 3D humans. The modality gap between RGB renderings and human shapes leads to different modeling between the two tasks. Therefore, the objective of our approach has a significant difference compared with gDNA. Moreover, we have cited and properly discussed gDNA in Section 2 of our paper (L88) as part of 3D human generation, where we believe gDNA will shed light on 3D human generative model on modeling more detailed geometry.
>
> **Q6: Experiments on THuman dataset.**
>
> Thanks for your constructive comments. We train our model on THuman 2.0 dataset. Overall, we render 500 identities from THuman 2.0 with 36 camera views for each identity. We keep the training configuration unchanged and retrain our model from scratch on the rendered images. Note that, no explicit 3D supervision, e.g., normal or mesh, is used in our training. We present the results in the **Figure 2 of the attached figure-only file**, where the renderings show promising results.
>
> **Q7: Provide analysis for the failure of the voxel-based approach.**
>
> Thanks for your question. The voxel-based approach densely allocates voxel grids with a resolution of $64^3$ to model 3D humans. As discussed in L263-L265, it wastes lots of parameters to densely model the entire scene instead of the human body itself, leading to the low spatial resolution of the representation. This is the root cause of its blurry renderings, which validates the importance of primitive-based representation for 3D human diffusion model.

---

> > ### Comment · Reviewer_GMsy · 2023-08-15
> >
> > Dear authors,
> >
> > thanks for the detailed rebuttals. I don't have additional questions and will make a final decision based on the discussion with other reviewers.
> >
> > Best,

---

> > > ### Author Response · Authors · 2023-08-16
> > >
> > > Thanks for your feedback. We are encouraged to see that our responses address your concerns above. In addition, we would like to emphasize our contributions as follows:
> > > 1) We propose the first diffusion model for 3D human generation.
> > > 2) We design a compact and expressive parameter space on top of volumetric primitives to enable tractable training of diffusion models and high-performance rendering in a decoder-free manner.
> > > 3) We propose a scalable framework with cross-modal attention for learning volumetric primitives without per-subject optimization on large-scale dataset.
> > > 4) We demonstrate the great potential of our work in training-free generative tasks like texture transfer and 3D inpainting, which surpasses the limitations of prior approaches that necessitate retraining.
> > >
> > > Should there be any further concerns from you that might hinder the recommendation of our paper for acceptance, we would be glad in addressing them.

---

### Official Review · Reviewer_aNCP · 2023-07-01

**Soundness:** 3 good
**Presentation:** 3 good
**Contribution:** 3 good
**Rating:** 6
**Confidence:** 4

**Summary:**

The paper proposes a generative model for 3D humans. The work is based on volumetric rendering of geometric primitives and diffusion model. Given multi-view images of humans as training data, the method first learns a volumetric primitives representation, and then applies diffusion model framework to model the representation distribution. The authors evaluate their method and compare to SOTA approaches on a relatively small synthetic multi-view dataset. Results show favor towards proposed approach.

**Strengths:**

1. The paper tackles an interesting, practical and high-impact problem: generating clothed 3D humans.

2. The paper is overall well written, easy to follow.

3. The use of cross attention to fuse pose info and texture info is appreciated and demonstrated good performance.

4. I'm glad to see the attempt of using volumetric primitives and diffusion model to create generative models of 3D humans.

**Weaknesses:**

1. The bibliography needs to be better formatted.
a) The same conference is referred in different format, e.g. CVPR vs IEEE Conference on Computer...
b) Capitalization of words, e.g. 3d -> 3D, Shapenet -> ShapeNet
c) Others, e.g. [30] two commas, [31] where is it published?

2. As a reference, I'd like to have a fidelity evaluation of "ground truth" volumetric primitives, better added into Table 1 in the updated version. This helps to understand the upper limit of the diffusion process in the proposed method and see where the bottle neck lies.

3. It looks like all the generated results are blurry. Is it due to the memory limit? What is the GPU memory consumption for W=32 (assuming it is the setting of all the experiments)? Is the memory consumption a limitation for the proposed method compared to EVA3D? In other words, if one had a large size of multi-view human data, would it be likely that EVA3D performs much better than the proposed method given current limit of computational power?

4. Most of the generated results presented in the submission (including the video) contain mono-color clothing. There are only two or three exceptions. Is it a bias of the dataset or is it a result of low resolution volumetric primitives? Could the authors provide a statistics on the ratio of mono-color clothing in the Renderpeople dataset?

5. What is the difference of texture transfer and reposing? It looks to me that transfer the source texture to the target subject is the same as repose (and reshape the body of) the target subject to the source pose (and source body shape). If this is true, please remove this application as it is basically in 4.4 already.

**Questions:**

I would like the authors to answer Weaknesses 2, 3, 4, and 5. The answers may not change my final decision as long as no major flaw is found later. The lack of details in the generated results prevent me from giving strong support of the submission. But the attempt to address 3D generative human models with diffusion model and volumetric primitives should be encouraged.

**Limitations:**

Yes, both are tackled in the submission.

---

> ### Author Rebuttal · Authors · 2023-08-09
>
> Thank you for your careful review and constructive suggestions. We summarize and answer your questions as follows:
>
> **Q1: The bibliography needs to be better formatted.**
>
> Thank you for the great suggestion. We will revise all references in a unified format.
>
> **Q2: Fidelity evaluations of ground truth volumetric primitives (Table 1) as an upper bound for diffusion.**
>
> Thanks for this constructive comment! We provide all corresponding metrics in Table 1 for ground truth volumetric primitives learned from the first stage as follows. As you correctly pointed out, this serves as the upper bound of our diffusion model. We will add these numerical results in Table 1 of our revised version.
>
>
> |Methods | FID$_{\mathrm{CLIP}} \downarrow$ | FID $\downarrow$ | KID$\times 10^2 \downarrow$ | PCK $\uparrow$ | Depth$\times 10^2 \downarrow$|
> |:------:|:------:|:------:|:------:|:------:|:------:|
> |GT|10.34|12.66|1.67 $\pm$ 0.10|97.70|1.20 $\pm$ 1.65|
> |Ours|12.11|17.95|1.63 $\pm$ 0.09|97.62|1.42 $\pm$ 1.78|
>
>
> **Q3: Are blurry results caused by GPU memory limit? Provide the GPU memory consumption for W=32.**
>
> Thanks for your question. We first provide the GPU memory usage during training and inference of the diffusion model. Please kindly refer to the above post named "Response to the common issue of the computational cost". Ideally, we could have more detailed results by increasing the resolution of W. We observe that one of the reasons for some blurry results is that we remove the view condition from the model to force it to learn a truly 3D representation. As shown in Figure 1 of the supplementary material and Section B.3, the UV-aligned view conditions help volumetric primitives to capture more details. However, such a way of conditioning makes the model relies on queries on the view decoder at inference time. It breaks the principle of decoder-free rendering, leading to slower rendering speed for novel poses and views.
>
> **Q4: The statistics of mono-color clothing in the dataset.**
>
> Thanks for your question. We provide the statistics of mono-color clothing in our dataset, which shows a strong bias towards mono-color clothes. Therefore, the results are caused by dataset bias. In the future, one may seek larger scale data with more diversity for training or use ControlNet [1] to augment the dataset.
>
> | Body Part| Mono-color   | Complex Texture |
> | :----: | :------: | :------: |
> | Upper (shirts, jacket etc.)| 698 | 98 |
> | Lower (pants, trousers etc.)| 754 | 42 |
>
> [1] L. Zhang et al. Adding Conditional Control to Text-to-Image Diffusion Models. In arXiv, 2023.
>
> **Q5: The difference of texture transfer and reposing.**
>
> Let us elaborate on the difference between texture transfer and pose-driven synthesis (reposing). They are only equivalent if we want to transfer all textures (shoes, shirts, and pants) from the source to the target. In most cases, the goal of texture transfer is to transfer a part of the textures from the source to the target, e.g., changing a shirt for the person. As shown in Figure 8 of the paper (changing shirt), we leverage the dense correspondence provided by volumetric primitives to transfer the shirt texture from the source to the target. Reposing and reshaping the target person to the source one cannot offer the flexibility for controlling specific body parts or clothing.

---

### Official Review · Reviewer_PWaT · 2023-07-04

**Soundness:** 3 good
**Presentation:** 3 good
**Contribution:** 3 good
**Rating:** 5
**Confidence:** 5

**Summary:**

The paper presents a diffusion-based approach, PrimDiffusion, for 3D human generation. The method leverages the representation of MVP (mixture of volumetric primitives) and a design of volumetric primitives diffusion to generate high-quality 3D human models using multi-view images as training data. The authors propose a two-stage training process: the first stage involves learning generalizable primitives from multi-view images, and the second stage involves a diffusion process to generate/sample these primitives. The paper also presents an encoder design tailored with cross-modal attention for learning volumetric primitives from images across identities. This encoder, which consists of an appearance branch and a motion branch, is trained end-to-end across identities, eliminating the need for per-subject optimization. The cross-modal attention module within the encoder effectively captures dependencies between appearance and motion, contributing to the generation of high-quality 3D humans. The paper includes extensive experimental results demonstrating the effectiveness of the proposed method in comparison with existing techniques.

**Strengths:**

(+) A novel way of generating 3D human models using volumetric primitives diffusion.

(+) The paper is well-structured and clearly written. The authors provide a clear explanation of their method. The supplementary document provides additional technical details, and the supplementary video also helps to showcase the quality that the proposed method achieves.

(+) The proposed method achieves superior results compared to existing techniques on the RenderPeople multiview dataset.

**Weaknesses:**

(-) The proposed method is a 2-stage approach, compared to previous GAN-based methods such as EVA3D, it requires **multi-view images** as training data to learn generalizable primitives in the first stage. This is a significant difference as collecting large-scale real-world multi-view human data can be challenging, while 2D image collections are relatively easier to gather. This distinction is not clearly articulated in the paper. Considering a single-stage solution might address this issue.

(-) The experiments lack a comparison with a recent baseline: Jiang, Suyi, Haoran Jiang, Ziyu Wang, Haimin Luo, Wenzheng Chen, and Lan Xu. "Humangen: Generating human radiance fields with explicit priors." In Proceedings of the IEEE/CVF Conference on Computer Vision and Pattern Recognition, pp. 12543-12554. 2023.

(-) Please refer to the questions section for more points.

**Questions:**

- In Figure 3, top-left, the input is denoted as "Body Model $M(\beta, \theta)$", but the figure appears to depict a detailed model rather than a SMPL model. Could the authors clarify this discrepancy?

- When projecting image features and pose features to the SMPL UV space, how are the **"empty"** UV regions handled? What are the initial values for the primitive positions corresponding to these empty regions?

- What is the memory consumption of the proposed method during training and testing?

- The training data includes 20 different human poses. How were these poses sampled? What criteria were used to ensure a comprehensive coverage of possible poses?

**Limitations:**

The proposed method requires the use of multi-view images as training data. This requirement, when compared to previous works that do not have such a constraint, should be discussed in detail as a limitation of the current approach.

---

> ### Author Rebuttal · Authors · 2023-08-09
>
> Thank you for your careful review and constructive suggestions. We summarize and answer your questions as follows:
>
> **Q1: The proposed method is a 2-stage approach that requires multi-view images.**
>
> Thanks for your feedback. Indeed, this is a limitation of our method which we will discuss in our revised version. Besides, as discussed in L311 of our paper, the current two-staged approach is a design choice for training stability given the high degrees of freedom of primitives. It is fruitful to explore a single-stage approach like SSDNeRF [1] for 3D humans in the future. Yet, efficient 3D representation for highly articulated objects like humans is still an open problem for single-staged methods. We believe our method paves the way for solving this challenging problem.
>
> **Q2: Lack a comparison with HumanGen.**
>
> Thanks for your constructive comments.
> HumanGen [3] is a great work for 3D human generation that leverages explicit image priors from strong pretrained 2D human image generators and geometry priors from PIFu [4]. Unfortunately, it is published in CVPR 2023 which was held after the submission date of NeurIPS. To the best of our knowledge, the source code has not been released so far. Therefore, we tried to reproduce according to our understanding. However, we failed to achieve reasonable results in our setting. Undoubtedly, we will cite and discuss HumanGen as a concurrent work in Section 2 of our revised version.
>
> **Q3: Clarification of Figure 3 top-left on why it is a more detailed model than SMPL.**
>
> Sorry for the confusion. The top-left of Figure 3 is only a principled illustration of parametric human models. It could be any parametric human model, like SMPL or SMPL-X. In our implementation, we choose SMPL. Will replace it with the reposed SMPL mesh for coherence in the revised version.
>
>
> **Q4: Memory consumption during training and testing.**
>
> Thanks for the question. Please kindly refer to the above post named "Response to the common issue of the computational cost".
>
> **Q5: How the 20 poses are sampled for training data?**
>
> Thanks for the question. As introduced in the supplementary material (Section C.2), we randomly sample human poses from AMASS dataset [2], which covers sufficient diversity of real-world human poses. Moreover, to bridge the gap between SMPL models from AMASS dataset and arbitrary human meshes from the RenderPeople data, we use the motion retargeting technique which allows for the transfer of motion data between heterogeneous sources.
>
> **Q6: How are the empty UV regions handled?**
>
> Let us elaborate on this process in detail. The pose features are padded to align the UV space so there is no projection process. We choose an "inverse" way to project the image features. Specifically, we project the vertices of the SMPL model to the corresponding conditioned camera view to get pixel coordinates and perform grid sampling on the conditioned 2D image to get the corresponding projected image as input. Since the UV mapping of the SMPL model is predefined, there is no need to handle empty UV space in this process. As you correctly pointed out, we need to handle empty UV regions to initialize the primitives' positions. As presented in L131 of our paper, The 3D positions of primitives are initialized by uniformly sampling UV space and mapping each primitive to the closest surface point on SMPL that corresponds to the UV coordinates of the grid points. Specifically, we handle empty UV regions by inpainting the vertices index map with the nearest neighbors in the UV space. Intuitively, the primitive initialized to those "empty" UV regions will be placed on the nearest SMPL vertex.
>
> [1] H. Chen et al. Single-Stage Diffusion NeRF: A Unified Approach to 3D Generation and Reconstruction. In ICCV, 2023.
>
> [2] N. Mahmood et al. AMASS: Archive of motion capture as surface shapes. In ICCV, 2019.
>
> [3] S. Jiang et al. HumanGen: Generating Human Radiance Fields with Explicit Priors. In CVPR, 2023.
>
> [4] S. Saito et al. PIFu: Pixel-Aligned Implicit Function for High-Resolution Clothed Human Digitization. In ICCV, 2019.

---

> > ### Comment · Reviewer_PWaT · 2023-08-17
> >
> > Dear Authors,
> >
> > Thank you for providing a rebuttal in response to the initial review. However, there remain some unresolved concerns:
> >
> > - For my "Q1: The proposed method is a 2-stage approach that requires multi-view images", the primary concern was the method's dependency on **multi-view images for training**. This was not explicitly addressed in the authors' response.
> >
> > - Regarding my "Q5: How the 20 poses sampled for training data?", employing a simple random sampling strategy may not yield a representative pose distribution from the AMASS dataset. Considering the **imbalanced motion distribution within AMASS**, this strategy might compromise the effectiveness of the proposed method. It would be beneficial if you could provide further insights or clarifications.
> >
> >
> > Best,
> >
> > Reviewer PWaT

---

> > > ### Author Response · Authors · 2023-08-19
> > >
> > > Thank you for your valuable feedback and follow-up comments. We appreciate the opportunity to provide additional clarifications in response to your concerns regarding Q1 and Q5.
> > >
> > > **Q1**: We acknowledge the consideration that multiview images incur higher costs compared to image collections. However, we argue that training with multiview images offers distinct advantages, namely 1) learning comprehensive and coherent 3D representation of human body (e.g., texture on the back of human body) and 2) enabling view synthesis with free viewpoints. Notably, existing approaches like ENARF-GAN [1] and EVA3D [2] opt for image collections for training due to their cost-effectiveness, yet they are constrained to generating forward-facing humans within limited camera view variations. In contrast, our model offers a promising way to fully utilize the multiview knowledge during training to generate coherent 3D representation of humans, enabling free-view synthesis and efficient decoder-free rendering. Moreover, the concurrent work HumanGen also achieved 360 degree rendering by incorporating explicit 3D priors from the model trained on multiview images. This substantiates the effectiveness and necessity of utilizing multiview data for comprehensive 3D human generation. In essence, both image collections and multiview data serve as distinct task settings tailored to different objectives. The former can leverage vast in-the-wild knowledge in a cheaper way but the camera views are bounded by the training data. The latter, albeit with a higher cost associated with multiview setup, facilitates the learning of genuinely 3D human representations, enabling free-view rendering.
> > >
> > > **Q5**: Thanks for your insightful comments. We note that the generalizability of our method to novel poses has been validated in our experiments (supplementary video) where all novel poses are out-of-distribution poses instead of in-distribution unseen poses. This feature is enabled by our careful design of decoder-free rendering, which exclude the implicit dependence of pose during rendering. Therefore, the diversity of human poses does not significantly impact our training process. Furthermore, we conducted experiments of three different sampling strategies on AMASS dataset. In these experiments, we sample $20\times 796 = 15920$ poses (equals to the number of poses used in our RenderPeople dataset) from AMASS for each method and compute the Fréchet distance [3] on the pose parameters of SMPL-X against the source data. It's important to emphasize that the Fréchet distance serves as a measure of distribution similarity. We investigate three different pose sampling strategies: 1) uniform: sample poses from the dataset with a fixed interval, 2) random: sample poses from the dataset randomly, and 3) cluster: we divide the dataset into 30 pose clusters using K-means algorithm, and randomly sample poses in each cluster. The results shown below indicate that the difference between sampling strategies is minor given the number of sampled poses we used for training is already large enough.
> > >
> > > |Methods|Fréchet distance ($\times 10^{-3}$)|
> > > |:--:|:--:|
> > > |uniform|0.99|
> > > |random|1.13|
> > > |cluster|4.48|
> > >
> > > [1] A. Noguchi et al. Unsupervised Learning of Efficient Geometry-Aware Neural Articulated Representations. In ECCV, 2022.
> > >
> > > [2] F. Hong et al. EVA3D: Compositional 3D Human Generation from 2D Image Collections. In ICLR, 2023.
> > >
> > > [3] D.C Dowson et al. The Fréchet distance between multivariate normal distributions. In Journal of Multivariate Analysis, 1982.

---

### Official Review · Reviewer_wB9H · 2023-07-06

**Soundness:** 3 good
**Presentation:** 3 good
**Contribution:** 2 fair
**Rating:** 5
**Confidence:** 5

**Summary:**

The paper presents, PrimDiffusion, an approach for generative modeling of 3D humans. PrimDiffusion is a diffusion-based model that operates on a set of volumetric primitives representing the human body. Each volumetric primitive consists of rotation, translation, scale, radiance, and color information. PrimDiffusion is a multi-stage approach. Given a training dataset with multiview images, the first stage fits volumetric primitives on training images using reconstruction losses. The second stage then uses the fitted volumetric primitives to train a diffusion-based generative model where the results from the first stage are used as the ground truth for the diffusion process.  The use of volumetric primitives allows decoder-free rendering of the images which allows real-time rendering once the diffusion process is done. It also provides explicit control over the body pose and viewpoint. The paper also shows its application for texture transfer and unconditional infilling. Experiments are performed on the synthetic RenderPeople dataset where PrimDiffusion is shown to outperform other baselines.

**Strengths:**

- The paper addresses the challenging problem of generative modeling of 3D humans.
- The use of volumetric primitives to represent complex human bodies, clothing, and hairs makes sense and is intuitive.
- The main challenge for learning 3D diffusion models is the availability of 3D ground truths. The paper overcomes this challenge by first fitting volumetric primitives on multiview images.
- The paper inherits the advantages of volumetric primitives including decoder-free rendering and explicit control over viewpoint and body pose.
- The paper demonstrates two interesting applications of the proposed method including texture transfer and infilling.
- The paper is well-written and easy to read.

**Weaknesses:**

### Novelty
- The main novelty of the proposed method comes from the combination of existing bits and pieces. The volumetric primitives have already been used in [25] to model human faces. The paper extends it to the full body by incorporating the SMPL body model. The idea of anchoring neural primitives to the SMPL body model is similar to NeuralBody [a] `(missing citation)`, and generative modeling using diffusion models is standard practice these days. While there is no technical novel contribution of the paper, I acknowledge that the proposed combination is not straightforward and this is not the most crucial limitation of the paper.

### Comparison with existing methods
- This is the most crucial limitation of the paper. The paper compares with EVA3D, StyleSDF, and EG3D.  The main advantage of these methods is that they can learn 3D representation from a large collection of 2D images. However, it seems that this paper has retrained them from scratch using only the multi-view images from the RenderPeople dataset `(please correct if my understanding is wrong)`. Hence, their results are significantly worse than what is shown in the original paper, in particular the results of EVA3D. I find this comparison unfair for these methods since it suppresses their advantages and conceals the limitations of the proposed method. For a fair comparison, these methods should be fine-tuned on the RenderPeople dataset rather than training from scratch. It is the limitation of the proposed method that it requires 3D ground truths for training and hence can only be trained on multiview data. Other methods, on the other hand, can use a large collection of 2D images hence they should be used by default for these methods.

### Blurry results
- The qualitative results are quite blurry. Would be great if the authors can comment on this during the rebuttal.

### Missing references
[a] Implicit Neural Representations with Structured Latent Codes, CVPR'21

### Conclusion
Overall, it is a good paper with a reasonable approach. However, the experiments section of the paper is a bit weak, in particular, the comparison with existing methods is unfair. Hence, my current rating for the paper is `Weak Reject`. I would be happy to upgrade my rating if the authors can address my concerns during the rebuttal.

**Questions:**

### Missing Experiments
- The paper learns 3D primitives using reconstruction losses and multiview images. Since the ground-truth meshes are available for the RenderPeople dataset, an interesting comparison would be to use them to obtain volumetric primitives. It will help us understand how much information is lost during the learning of volumetric primitives using multi-view images.
- I would have liked to see a baseline that is similar to NeuralBody i.e., replace volumetric primitives with learnable latent codes.
- Please explain how the baseline methods are trained.

**Limitations:**

- As mentioned above, reliance on 3D ground-truths is a significant limitation of the proposed method as compared to existing methods. Hence, it should be discussed in the `Limitations`.

---

> ### Author Rebuttal · Authors · 2023-08-09
>
> Thank you for your careful review and constructive suggestions. We summarize and answer your questions as follows:
>
> **Q1: Novelty of the proposed method.**
>
> Thanks for your comments. We highlight our key contributions in the above post named "Response to the common issue of the novelty and motivation".
>
> Moreover, we argue that our human representation is fundamentally different from NeuralBody. **First**, we anchor explicit primitives with color and kinematic information to the human model while NeuralBody leverages implicit latent codes. **Second**, NeuralBody relies on decoders (consisting of sparse convolution layers and MLPs) to 1) diffuse latent features off the surface of SMPL topology and 2) render latent features into color and density. It provides no generalizability to novel poses as the decoder overfits to the training observations. Instead, the volumetric primitive in our case is decoder-free, which enables efficient rendering and generalizing to novel poses. Moreover, our model is capable of handling off-body topology like loose garments while the latent codes of NeuralBody cannot.
>
>
> **Q2: Unfair comparisons for baseline methods.**
>
> Thanks for your great suggestion. We finetuned EVA3D which is pretrained on DeepFashion dataset according to your suggestion. The numerical results are summarized below. We found that it indeed provides a slight improvement compared with the one trained from scratch. However, it still has a significant gap compared with our method. Moreover, we present qualitative results in **Figure 1 of the attached figure-only file**. Will add this result in Table 1 and Figure 5 of our revised version.
>
> Besides, we argue that stage 1 of our method can be treated as an inverse rendering step that recovers 3D representation from 2D observations. It can further benefit from techniques like sparse views reconstruction which can reduce the reliance on multiview images. The key insight of our paper is proposing a way to inherit both the capacity of the diffusion model and the efficiency of volumetric primitives.
>
> |Methods | FID$_{\mathrm{CLIP}} \downarrow$ | FID $\downarrow$ | KID$\times 10^2 \downarrow$ | PCK$\uparrow$ | Depth$\times 10^2 \downarrow$|
> |:------:|:------:|:------:|:------:|:------:|:------:|
> |EVA3D|15.03|44.37|2.68 $\pm$ 0.13|91.84|3.24 $\pm$ 9.93|
> |EVA3D (Finetuned)|14.58|40.40|2.99 $\pm$ 0.14|91.30|2.60 $\pm$ 7.95|
> |Ours|12.11|17.95|1.63 $\pm$ 0.09|97.62|1.42 $\pm$ 1.78|
>
>
> **Q3: Blurry results.**
>
> Thanks for your comments. We observe that one of the reasons for some blurry results is that we remove the view condition from the model to force it to learn a truly 3D representation. As shown in Figure 1 and Section B.3 of the supplementary material, the UV-aligned view conditions help volumetric primitives to capture more details. However, such a way of conditioning makes the model relies on queries on the view decoder at inference time. It breaks the principle of decoder-free rendering, leading to slower rendering speed for novel poses and views.
>
>
> **Q4: Using GT mesh from RenderPeople to obtain volumetric primitives.**
>
> This is a good point! In the context of single identity reconstruction, the ground truth mesh undoubtedly offers better geometry than the estimated SMPL model, resulting in a more accurate initialization of rotation and translation for each primitive.
> However, as a generative model, our objective is not solely focused on precisely reconstructing a single identity. Instead, we aim to establish a unified parameter space that can be shared across identities for effective representation learning by diffusion model. The ground truth meshes obtained from the RenderPeople dataset contain varying numbers of vertices, preventing the establishment of a unified UV space and primitives parameter space suitable for training.
>
> **Q5: A baseline similar to Neuralbody, replacing volumetric primitives with latent codes.**
>
> Thanks for your suggestion. Note that, NeuralBody is an approach for single identity reconstruction while our method aims at human generation. It takes 14 hours per subject to converge which makes it unscalable to large-scale datasets like RenderPeople. Thus, we replace the payload of volumetric primitives to latent codes according to your suggestion and leverage sparse convolution layers as the decoder to output color and density. All kinematic parameters of primitives are frozen. This implementation of a NeuralBody version of our method keeps the ability to learn cross-identity 3D representation, getting rid of per-subject fitting. We present quantitative results below and show qualitative results in the **Figure 3 of the attached figure-only file**. Our method outperforms the NeuralBody (latent codes) variant. We will add this baseline as additional ablation studies to Table 2 in our revised version.
>
> |Methods | FID $\downarrow$ | KID$\times 10^2 \downarrow$ | Depth$\times 10^2 \downarrow$|
> |:------:|:------:|:------:|:------:|
> |NeuralBody| 37.70 | 3.44 $\pm$ 0.11 | 1.76 $\pm$ 1.91|
> |Ours|17.95|1.63 $\pm$ 0.09|1.42 $\pm$ 1.78|

---

> > ### Comment · Reviewer_wB9H · 2023-08-15
> >
> > Dear Authors,
> >
> > Thank you for providing the fine-tuning experiment for EVA3D. It would be nice to provide further details about the training process. Is it trained only using the GAN losses? If so, I feel that it is still unfair to EVA3D since the proposed method heavily relies on 3D supervision whereas EVA3D is only provided with weak 2D supervision in the form of GAN loss, even the reconstruction loss is not used. I would have liked to see a baseline method that utilizes a similar amount of 3D supervision as the proposed approach.
> >
> > Regarding novelty, I acknowledge that the proposed combination is not straightforward and somewhat novel.
> >
> > Sincerely,
> > wB9H

---

> > > ### Author Response · Authors · 2023-08-16
> > >
> > > Thanks for your feeback and for acknowledging our technical contributions. We appreciate the opportunity to provide further clarification on the training process and address the concerns you raised.
> > >
> > > Regarding the training process, we would like to emphasize that our method does not require any explicit 3D supervision, such as ground-truth meshes or normals, during training. All methods in our experiments, are trained using a consistent experimental setup with input training data consisting of multiview 2D images and camera poses. This ensures that both EVA3D and our model, are provided with weak 2D image-based supervisions during training.
> > >
> > > The finetuned EVA3D is trained using the same configuration as the official code release. The training objective for the finetuned model involves a combination of non-saturating GAN loss, minimum offset loss, and eikonal loss. We utilized the pre-trained EVA3D model, which was initially trained on the DeepFashion dataset, and performed the finetuning on our RenderPeople dataset. During our experimentation, we observed that introducing a 2D image-based reconstruction loss to the training process of EVA3D led to mode collapse, likely due to the sensitivity of adversarial training. Too strong reconstruction loss will degenerate the model into a autodecoder.
> > >
> > > We also want to highlight that, as the first diffusion model for 3D human generation, direct comparisons with prior works in the exact same setting are challenging due to the novelty of our approach. However, to provide meaningful baselines for comparison, we included results from voxel-based and NeuralBody-based methods. These baselines utilize the same 2D image-based supervision as our approach, enabling a fair assessment of our method's performance.
> > >
> > > Furthermore, the comparison of 3D diffusion models with GAN-based methods on multiview images is a widely accepted practice in recent diffusion models for 3D rigid object generation [1,2,3]. We follow this practice for comparisons of 3D human (non-rigid body) generation, which allows for meaningful evaluations and insights into the capabilities of different methods in generating 3D representations from 2D inputs.
> > >
> > > We hope this clarification addresses your concerns and provides a more comprehensive understanding of our experiment and method. We are open to any further discussions or questions you may have, and we appreciate your consideration of our paper for acceptance.
> > >
> > > [1] N. Muller et al. DiffRF: Rendering-Guided 3D Radiance Field Diffusion. In CVPR, 2023.
> > >
> > > [2] T. Wang et al. Rodin: A Generative Model for Sculpting 3D Digital Avatars Using Diffusion. In CVPR, 2023.
> > >
> > > [3] E. Ntavelis et al. Autodecoding Latent 3D Diffusion Models. In arXiv, 2023.

---

> > > > ### Comment · Reviewer_wB9H · 2023-08-22
> > > >
> > > > Thank you for providing a detailed response. What I meant was that EVA3D can probably benefit from adding multiview consistency losses, in addition to the GAN loss that it currently uses for monocular images. Nonetheless, most of my concerns have been addressed. Thank you!

---

### Official Review · Reviewer_oDDi · 2023-07-10

**Soundness:** 3 good
**Presentation:** 3 good
**Contribution:** 4 excellent
**Rating:** 6
**Confidence:** 4

**Summary:**

The paper introduces a novel framework called PrimDiffusion, which utilizes diffusion models to generate 3D human models. The framework leverages a Mixture of Volumetric Primitives (MVP) to represent 3D humans, enabling efficient rendering and incorporating human priors for retargeting. This representation offers a compact and expressive 2D parameter space, making it well-suited for the diffusion operation. Experimental results demonstrate that the proposed model outperforms previous approaches in terms of both quality and efficiency. Furthermore, the proposed method exhibits flexibility in training-free conditional generation tasks, including texture transfer and 3D inpainting.


**Strengths:**

The paper clearly proposes to represent 3D humans using volumetric primitives, effectively transforming the 3D representation into a 2D parameter space that is well-suited for 2D diffusion. This method outperforms previous approaches for 3D human face/body generation, which mostly rely on triplane representation. Further, volumetric primitives technique also provides easier animation and conditional generation capabilities.

Through experiments, the proposed method demonstrates superior qualitative and quantitative performance compared to previous works. These experiments highlight the advantages of using volumetric primitives representation for 3D human generation, specifically, the paper compared it with voxel-based representations.


**Weaknesses:**

In lines 139-140, the proposed method only considers color, density, and delta scale factor as optimized parameters, while delta translation and delta rotation are not included. It would be beneficial to understand the reasons behind this omission, especially considering that the original MVP paper optimized these parameters.

Figure 2 illustrates the diffusion process of rotation (R), translation (T), and scale (s) during training, which appears to contradict the claim that the learnable parameters are solely color, density, and delta scale factor. And it seems that there is a discrepancy, as the expected parameters should be delta rotation, delta translation, and delta scale. Clarification on this point would be appreciated.

The motion encoder in the proposed method solely considers joint rotations, which may not be sufficient for accurate fitting of 3D humans. Shape parameters also play a crucial role in this process. It would be helpful to hear an explanation for the decision to exclude shape parameters from the motion encoder.

There are typos in Function (7), which are inconsistent with the function presented in the DDPM paper [16]. It is important to address these inconsistencies and ensure the correctness and coherence of the equations.

The loss function (8) differs from the one described in the DDPM paper [16]. The original function aimed to estimate noise from V_t instead of directly estimating the ground truth. It would be beneficial to provide an explanation for this modification and discuss its implications.


**Questions:**

The paper conducted a comparison between volumetric primitives and voxel representations in diffusion-based 3D human generation. However, I am also interested in understanding how volumetric primitives compare to triplane representations in diffusion. It would greatly enhance the experimental analysis if this comparison were included.

Regarding the ablation study, the paper compared predefined rotation and translation values derived from a template human mesh with identity-specific ones. However, the paper does not provide a detailed explanation of the identity-specific rotation and translation. It would be beneficial to include a thorough explanation of this approach to improve the understanding of the study.


**Limitations:**

The proposed method is innovative as it combines volumetric primitives with diffusion model. However, there is room for improvement in terms of the writing. There is still a noticeable disparity between the ground truth and the fitting results. It seems that there is significant room for improvement in stage 1 of the method.

---

> ### Author Rebuttal · Authors · 2023-08-09
>
> Thank you for your careful review and constructive suggestions. We summarize and answer your questions as follows:
>
> **Q1: The reasons behind the omission of delta translation and delta rotation.**
>
> Thanks for your comments. There are two reasons for predicting delta scale factor only: 1) better generalizability to novel poses, 2) efficient decoder-free rendering. We empirically found that it is only beneficial to include delta translation and rotation for reconstruction quality. However, it does harm to the generalizability of novel poses as the model now implicitly depends on training pose distribution. When we drive the primitives with out-of-distribution poses once the denoising process is done, it requires queries of delta translation and rotation decoder. However, such a decoder-dependent pipeline prevents the model from generalizing to novel poses and slows the rendering speed.
>
> **Q2: Clarification on the discrepancy between Figure 2 and text on whether or not to optimize rotation(R), translation(T).**
>
> Thanks for your careful reading and outstanding review! The $R_t$ and $T_t$ should always equal $R_0$ and $T_0$. We will clarify this to be consistent with the caption of Figure 2 in the revised version.
>
> **Q3: The reason for excluding shape parameter from motion encoder.**
>
> Thank you. The reason for excluding the shape parameter from the motion encoder is that the shape parameter has been explicitly considered through the process of linear blending skinning, which takes as input the pose $\theta$, shape $\beta$, and the neutral SMPL model template.
>
> **Q4: Clarifications of Equation (7) and (8).**
>
> Thank you for the comment. We empirically found that predicting $\mathcal{V}_0$ offers better convergence of the denoiser compared with predicting the noise $\epsilon$ in our case, which is also observed in other 3D diffusion models [1]. However, the formulation presented in the original DDPM paper is predicting the noise itself (Eq. 11 and Eq. 14 in their paper). Thus, we derive our formulation (Eq. 7 and Eq. 8) from the original one while keeping them mathematically equivalent.
>
> **Q5: Volumetric diffusion v.s. triplane diffusion?**
>
> This is a great point. In fact, our initial attempt started from triplane diffusion which is similar to Rodin [2]. However, we found that it is not straightforward to fit triplanes and train the diffusion model on top of it for 3D human bodies. As shown in **Figure 4 of the attached figure-only file**, triplane diffusion can easily generate floating artifacts around the human body. We attribute the failure of triplane diffusion on 3D humans to the inefficiency of triplane in representing the human body which is a highly articulated object. The human body only occupies a small portion of the space modeled by the triplane, thus the network wastes most of the parameters in modeling empty space. It poses challenges for both the fitting of triplanes and the convergence of diffusion models.
>
> **Q6: Detailed explanation of the difference between predefined rotation and translation and an identity-specific one.**
>
> Sorry for the confusion. Conceptually, the translation of each volumetric primitive is defined according to the positions of mesh vertices. And the rotation is defined in the TBN space over the human mesh. For the predefined rotation and translation, we compute them over a human mesh template shared across identities. For the identity-specific one, we compute the rotation and translation on the human mesh after the linear blending skinning with personalized shape and pose parameters. The intuition for this ablation is to evaluate whether or not using the delta scale factor only is enough to compensate for the inaccuracy of rotation and translation.
>
> [1] T. Anciukevicius et al. RenderDiffusion: Image Diffusion for 3D Reconstruction, Inpainting and Generation. In CVPR, 2023.
>
> [2] T. Wang et al. Rodin: A Generative Model for Sculpting 3D Digital Avatars Using Diffusion. In CVPR, 2023.

---

### Author Rebuttal · Authors · 2023-08-09

**Common response**

We sincerely thank all reviewers for the constructive comments and recognition of our work. We are encouraged that reviewers find our proposed method "innovative" (Reviewer oDDi), "novel" (Reviewer PWaT), and "effective" (Reviewer GMsy); our results "have superior performance" (Reviewer oDDi, PWaT) and "interesting" (Reviewer wB9H, aNCP). We have posted responses for each reviewer respectively.

**Response to the common issue of the novelty and motivation**

Both Reviewer wB9H and GMsy are concerned about the novelty and motivation of our method. Admittedly, the volumetric primitives, for their efficiency and effectiveness, have been proposed to model human faces. Nevertheless, it is not trivial to make it work on high-resolution and efficient 3D human generation. Specifically, the idea of volumetric primitives is proposed for identity-specific 3D reconstruction. However, it is intractable to fit every single human given a large-scale 3D human dataset. Instead, we propose a scalable framework with cross-modal attention for learning volumetric primitives without per-subject optimization. Moreover, our work first shows how to unleash the power of the diffusion model on 3D human generation while keeping real-time performance for novel pose and novel view synthesis. It is not a simple application of the diffusion model on volumetric primitives. Instead, we carefully choose the target domain for the denoiser as the color, density, and delta scale factor. It facilitates decoder-free rendering, which is critical for efficiency and generalizability to novel poses.

Moreover, we summarize the motivations for leveraging volumetric primitives for 3D human generation compared to other 3D representations as follows:

**Flexibility and Adaptability**: Volumetric primitives offer a versatile and adaptive way to represent the articulated human body. By decomposing the human body into a collection of volumetric primitives, it can efficiently handle diverse poses and articulations within a compact parameter space. The effectiveness of offering compact parameter space for 3D generative model is also validated in EVA3D, where the authors propose a compositional representation to achieve a high-resolution 3D human generative model without the super-resolution module. This adaptability is challenging to achieve using other representations like point clouds and occupancy grids.

**Expressiveness**: The volumetric primitive representation allows for capturing the articulated structure of the human body by incorporating parametric human prior. Unlike point clouds, which lack explicit surface representations and may lead to loss of information, volumetric primitives provide a dense and detailed representation of the human body. Moreover, the volumetric primitive offers dense 3D correspondence of the human body, facilitating flexible controls on downstream generative tasks such as 3D in-painting and texture transfer. Other representations like point clouds, SDF, and occupancy grids do not have such flexibility.

**Efficiency and Real-time Interactivity**: The efficiency of volumetric primitive representation contributes to real-time capabilities, making it suitable for interactive applications and systems that require quick response times. Note that, we only claim the real-time performance for novel view and novel pose synthesis once the denoising process is done (as introduced in L60 of the paper). Besides, it enables tractable training with a heavy denoiser backbone to unleash the capacity of the diffusion model and the high performance of volumetric primitives at the same time.

**Response to the common issue of the computational cost**

In response to the concerns from Reviewer PWaT, aNCP, and GMsy, we report the computational cost of our model. First, we provide the GPU memory consumption for both training and testing of the denoiser as follows. We will add this table to the supplementary material in our revised version.

| |GPU Memory (GB)| Batch size |
| :----: | :----: | :------: |
| Training | 27428 | 4 |
| Inference | 21258 | 1 |

Moreover, we report the computational performance in light of the whole pipeline at inference time as follows. The FPS stands for novel view and pose synthesis of 300 frames. The amortized FPS indicates the FPS by considering the time of the denoising process. Our average inference FPS still outperforms baselines (the best baseline is only 22.97). More importantly, we only claim the real-time performance for novel view and novel pose synthesis once the denoising process is done (as introduced in L60 of the paper). In most cases, the generative backbones account for identity-specific information. Since we disentangle the pose and view control from the generative backbone in a physically explicit way, we only need to call the denoiser once for identity-specific appearance. Thanks to our decoder-free rendering, we do not need any forward pass through the denoiser for novel view and novel pose synthesis, which is the fundamental reason for real-time rendering. However, existing 3D generative models like EG3D, EVA3D, and StyleSDF implicitly condition view and pose as input features, which forces them to call the forward pass of heavy generative backbones upon the view and pose changes. We will add this table to the supplementary material in our revised version.

| Inference Mode| Denoising time | FPS | Amortized FPS |
| :----: | :----: | :------: | :------: |
| DDIM, 100 steps | 2.86 s | 88.24 | 47.93 |
| DDIM, 50 steps | 1.45 s | 88.24 | 61.86 |

Please do not hesitate to let us know if you have any additional comments or if there are more clarifications that we can offer.

---

> ### Comment · Reviewer_aNCP · 2023-08-18
> **Typo in GPU Memory**
>
> Thank you very much for the rebuttal and clearance of most the questions.
> It seems the GPU Memory consumption in the above table has wrong unit. Instead of 20K+ GB it should be 20K+ MB. Please note that when adding it into revised version.

---

> > ### Author Response · Authors · 2023-08-19
> >
> > Thank you for your feedback. We appreciate your attention to detail. In the revised version, we will ensure that the GPU Memory consumption unit in the table is corrected to 'MB' and will include it in the supplementary material.

---

### Decision · Program_Chairs · 2023-09-21

**Decision:**

Accept (poster)

**Comment:**

The manuscript received two weak accepts, two borderline accepts, and one borderline reject. It combines the diffusion model with volumetric primitives to facilitate 3D human generation. A common concern among reviewers was the experimental comparison, which the authors addressed in their rebuttal. Although the generated results tend to be smooth, the AC also supports the effort to use volumetric primitives and the diffusion model in developing generative models and, therefore, recommends acceptance.